# Comprehensive Comparison of Effects of Antioxidant (Astaxanthin) Supplementation from Different Sources in *Haliotis discus hannai* Diet

**DOI:** 10.3390/antiox12081641

**Published:** 2023-08-19

**Authors:** Weiguang Zou, Jiawei Hong, Wenchao Yu, Yaobin Ma, Jiacheng Gan, Yanbo Liu, Xuan Luo, Weiwei You, Caihuan Ke

**Affiliations:** 1State Key Laboratory of Marine Environmental Science, College of Ocean and Earth Sciences, Xiamen University, Xiamen 361102, China; wgzou1994@stu.xmu.edu.cn (W.Z.); hongjiawei@stu.xmu.edu.cn (J.H.); yuwenchao@stu.xmu.edu.cn (W.Y.); yaobinma@stu.xmu.edu.cn (Y.M.); jiachenggan@stu.xmu.edu.cn (J.G.); 22320221151377@stu.xmu.edu.cn (Y.L.); wwyou@xmu.edu.cn (W.Y.); 2State Key Laboratory of Mariculture Breeding, Xiamen University, Xiamen 361102, China; 3National Observation and Research Station for the Taiwan Strait Marine Ecosystem, Xiamen University, Zhangzhou 363400, China

**Keywords:** antioxidant, astaxanthin, *Haliotis discus hannai*, diet supplementation

## Abstract

Dietary antioxidant supplementation, especially astaxanthin, has shown great results on reproductive aspects, egg quality, growth, survival, immunity, stress tolerance, and disease resistance in aquatic animals. However, the effects of dietary astaxanthin supplementation from different sources are still unknown. A comprehensive comparison of survival, growth, immune response, antioxidant activity, thermal resistance, disease resistance, and intestinal microbial structure was conducted in dietary antioxidant supplementation from the sources of *Gracilaria lemaneiformis* (GL), industrial synthetic astaxanthin (80 mg/kg astaxanthin actual weight, named as group ‘SA80’), *Phaffia rhodozyma* (80 mg/kg astaxanthin actual weight, named as group ‘PR80’) and *Haematococcus pluvialis* (120 mg/kg astaxanthin actual weight, named as group ‘HP120’) at their optimal supplementation amounts. Furthermore, the SA80, PR80, and HP120 groups performed better in all aspects, including survival, growth, immune response, antioxidant activity, thermal resistance, and disease resistance, compared with the GL group. The PR80 and HP120 group also had a better growth performance than the SA80 group. In terms of heat stress and bacterial challenge, abalone in the PR80 group showed the strongest resistance. Overall, 80 mg/kg astaxanthin supplementation from *Phaffia rhodozyma* was recommended to obtain a more effective and comprehensive outcome. This study contributes to the discovery of the optimum dietary astaxanthin supplementation source for abalone, which is helpful to improve the production efficiency and economic benefits of abalone. Future research can further explore the action mechanism and the method of application of astaxanthin to better exploit its antioxidant role.

## 1. Introduction

The excessive production of reactive oxygen species (ROS) caused by a loss of cell redox balance events can cause damage to DNA, proteins, lipids, and carbohydrates of the organism, finally resulting in cell death and tissue damage. It is well-established that antioxidants can inhibit the production of free radicals and eliminate the impairment of oxidative stress [1]. Except for endogenous antioxidants, there are exogenous antioxidants sources obtained from dietary supplementation, consisting of vitamins (vitamin A, vitamin C, and vitamin E), polyphenols, carotenoids, and oil lecithins, etc., which can serve as a good aids and complements [2]. Currently, dietary antioxidant supplementation is a hot topic in aquaculture: many related reports have been published on some aquatic species, like the greater amberjack (*Seriola dumerili*, curcumin supplementation) [3], whiteleg shrimp (*Litopenaeus vannamei*, vitamin E supplementation) [4], red sea bream (*Pagrus major*, selenium nanoparticle supplementation) [5], and Chinese mitten crab (*Eriocheir sinensis*, zinc supplementation) [6]. The addition of antioxidants to the diet is proven to be an effective method to improve animal immunity and growth performance.

Carotenoid is a kind of natural functional pigment that has been used as a dietary antioxidant supplement for aquaculture, playing other important biological functions besides colorants [7,8,9]. Among the carotenoids, astaxanthin (3,3′-dihydroxy-β,β-carotene-4,4′-dione) is the most representative, and is widely distributed in microorganisms and marine organisms [10]. Astaxanthin is known for its extraordinary antioxidative activity, which far exceeds other antioxidants like β-carotene, vitamin E, and α-tocopherol [11,12]. It has been intensively reported that astaxanthin shows great performance in reproductive health, egg quality, growth, survival, immunity, stress tolerance, and disease resistance in aquatic animals [13,14]. With multiple biological functions without the cost of adverse effects or toxicity making it favorable in the market, astaxanthin is also considered an eco-friendly functional feed additive in the aquaculture industry [15,16]. Being strongly attracted by the market demand and ecological benefit, practical applications and academic research related to dietary astaxanthin supplementation in aquaculture have become the focus recently.

Pacific abalone*, Haliotis discus hannai*, is a common economic shellfish that possesses a high nutritional and medicinal value [17,18]. It is the most populous species of farmed abalone around the world and is mostly cultured in China. By 2020, the abalone culturing yield of China exceeded 200,000 metric tons, accounting for about 90% of the global production [19,20]. It is well known that temperature is a limiting factor to pacific abalone: at a high level (above 28 °C), heat stress can cause functional problems, such as muscle neuronal migration abnormalities, mitochondrial disorder, and a decline in immunity and in the anti-oxidative capacity of abalone [21,22,23]. A crisis has arisen as it was reported that frequent massive mortality events were recorded because of high water temperatures in summer [24,25,26]. This will cause direct economic losses to a vast number of farmers, which restricts the development of the abalone farming industry.

Dietary antioxidant supplementation, especially astaxanthin, could be a gateway to solutions or mitigation, ensuring smooth production and output increase through use of its excellent biological functions. So far, a few researchers have attempted to investigate this. Lim and Lee [27] first added the yeast astaxanthin at a proportion of 1% in the diet of *H. discus hannai*; a significant growth advantage was not found in the results compared with other extracts within the group, with only an observable influence on shell color. The study of Ma et al. [28] made another attempt with a lower content (80 mg/kg, 0.8‱). There was a similar phenomenon to that which occurred in the growth performance with the earlier research, but an increase in antioxidative capacity and heat resistance was also observed. Since then, no more research has been conducted on dietary astaxanthin supplementation in Pacific abalone. There are still many key issues to be solved. 

The market is currently mainly dominated by industrial synthetic astaxanthin; however, with an increase in the rapidly growing demand, the green microalgae *H. pluvialis* and the red yeast *P. rhodozyma* are economical and efficient alternative sources of astaxanthin [29,30]. Therefore, researchers wish to elucidate the effects of dietary astaxanthin supplementation from natural and synthetic sources. This is the reason why we conducted this study. First, we addressed the question about the optimal addition content of astaxanthin from different sources through our previous experiments (results have not been published). And then, a comparison of the effects of dietary astaxanthin supplementation from different sources at their corresponding optimal quantities on pacific abalone was performed. Our study will analyze this through the aspects of growth performance, immunity, antioxidative capacity, heat resistance, disease resistance, and the intestinal microbial ecosystem. The results will give guidance on the application of dietary astaxanthin supplementation in the production, assisting in improving culturing and breeding efficiency, and enabling farmers to maximize the economic benefits of abalone.

## 2. Materials and Methods

### 2.1. Experimental Animals

The experimental abalones (*H. discus hannai*) were all provided by Fuda Abalone Aquafarm Co., Ltd. (Fuzhou, China). All abalones came from the same batch of artificial hatching and were later raised in the same culturing environment. Prior to the experiment, they were transferred to a cement pond (8 m × 2.2 m × 0.8 m) and acclimated for 15 days to adapt to the experimental diets. The pond was a flowing water system and was aerated all the time, and two-thirds of the seawater was replaced once a day to maintain good water quality. The water quality was controlled under the following conditions: temperature 22.5–23.5 °C, dissolved oxygen 7.24–7.36 mg/L, salinity 31–32‰, and pH 7.6–8.0, respectively.

### 2.2. Feeding Experiment

After acclimation, a total of 480 two-year-old abalones (initial shell length: 59.15 ± 0.63 mm; initial body weight: 24.53 ± 0.39 g) were randomly selected and divided into 4 diet groups (6 cages per group, 20 abalones per cage). Experimental diets were fed by hand until apparent satiation once a day at 18:00, and the feeding trial lasted for 100 days. Meanwhile, mortality was recorded daily. Culture management and condition control were consistent with the acclimatization period protocol described in 2.1.

In the present study, four diet groups were designed, including three isonitrogenous and isolipidic (32% crude protein, 3.8% crude lipid) experimental diets and a natural diet, *Gracilaria lemaneiformis* (16.37% crude protein, 1.86% crude lipid, named as group ‘GL’). The astaxanthin supplemented in the three experimental diets was from three different sources, respectively: industrial synthetic astaxanthin (Carophyll Pink, 10% astaxanthin, DSM, France), *Phaffia rhodozyma* (astaxanthin content of dry matter 0.4%, Fujian Lifecome Biochemistry Co., Ltd., Nanping, China) and wall-broken *Haematococcus pluvialis* dry powder (astaxanthin content of dry matter 3%, Yunnan Kunming Biogenic Co., Ltd., Kunming, China). The addition amounts of each were chosen according to their respective optimal amounts based on previous pre-experiments (unpublished): 800 mg/kg industrial synthetic astaxanthin supplementation (80 mg/kg astaxanthin actual weight, named as group ‘SA80’), 20 g/kg *Phaffia rhodozyma* supplementation (80 mg/kg astaxanthin actual weight, named as group ‘PR 80’), 4 g/kg wall-broken *Haematococcus pluvialis* dry powder supplementation (120 mg/kg astaxanthin actual weight, named as group ‘HP120’).

All experimental diets were provided by Fuzhou Promarine Biotechnology Co., Ltd. (Fuzhou, China). Fish meal, soybean meal, and wheat gluten were used as the main protein sources. Soy lecithin was used as the main lipid source. Kelp powder, high-gluten flour, dextrin, and α-starch were used as the main carbohydrate sources. All ingredients were ultra-micronized and screened using a 100 µm mesh, and then mixed. Experimental diets sources were finally pressed into flakes (2 cm × 2 cm) using a tablet press and dried at 60 °C for 24 h. Dried flakes were stored at −20 °C. Detailed information on the diets is shown in Table 1.

### 2.3. Sampling and Analysis

#### 2.3.1. Proximate Composition Analysis of the Diets

Proximate compositions of the diets were analyzed according to the methods described by the Association of Official Analytical Chemists AOAC [31]. Crude protein content was determined based on the Kjeldahl method. The Soxhlet method was used to determine the crude lipid content. The combustion in a muffle furnace (TMF-80-10TP, Shanghai, China) at 550 °C for 6 h was performed to analyze the ash content. Moisture was determined through drying using an oven (DHG-9140A, Shanghai, China) at 105 °C until a constant weight was achieved.

#### 2.3.2. Sampling

After the trial, all abalones went through a 3-day starvation period to make their intestines empty. Individual quantity, size, and weight were determined first, and then, the survival rate (SR), daily increment in shell length (DISL), weight gain rate (WGR), specific growth rate (SGR), and visceral–somatic index (VIS) were calculated. The calculation formulas are as follows:Survival rate (%) = 100 × (final number of abalones/initial number of abalones)
Daily increment in shell length (µm/day) = (final shell length − initial shell length)/breeding test days
Weight gain rate (%) = 100 × (final weight − initial weight)/initial weight 
Specific growth rate (%/d) = 100 × (Ln final weight − Ln initial weight)/breeding test days
Visceral–somatic index (%) = 100 × (viscera weight/total body weight)

Twelve individuals were randomly selected from each group, and approximately 1 mL of hemolymph was collected from the foot muscle using a sterilized syringe. The hemolymph samples were immediately centrifuged (3000× *g*, 10 min) to obtain serum. Following this, hepatopancreas and intestines from the abalones were collected from the and frozen in liquid nitrogen. All serum and tissue samples were stored at −80 °C for further analysis.

#### 2.3.3. Serum Immune Index and Hepatopancreas Antioxidant Enzyme Activity Analysis

Hepatopancreas samples were weighed and homogenized in cold saline solution (0.86%) of nine-fold of their weight. The homogenate was centrifuged at 3000× *g* for 10 min for enzyme activity analysis.

All the physiological and biochemical indexes were determined using commercial assay kits (Nanjing Jiancheng Bioengineering Institute, Nanjing, China) according to the manufacturer’s instructions. In detail, the glucose (GLU) and cortisol (COR) contents and alkaline phosphatase (AKP) and lysozyme (LZM) activity of the serum were analyzed using the corresponding kits (item number: A154-2-1, H094-1-1, A059-1-1, A050-1-1). Total antioxidant capacity (T-AOC), catalase (CAT), superoxide dismutase (SOD), and glutathione peroxidase (GSH-PX) activity, and the malondialdehyde (MDA) content of the hepatopancreas, were analyzed using the corresponding kits (item number: A015-2-1, A007-1-1, A001-3, A005-1, A003-1).

### 2.4. Heat-Resistance Evaluation of the Experiment

At the end of the feeding trial, fourteen abalones were selected from each group for heat-resistance evaluation. The heat resistance of abalone was evaluated by the heat adhesion duration (HAD) proposed by Yu et al. [32]; a lesser heat adhesion duration represents poorer heat resistance. In detail, abalones were labeled and transferred to a thermostatic glass tank (70 cm × 40 cm × 40 cm) with an adhesive substrate (45 cm × 25 cm) for abalone adhesion, and then acclimated at 20 °C for 7 days. After acclimation, the adhesion substrate with abalone attached was suspended vertically underwater, and the water temperature gradually increased from 20 °C to 33 °C at a rate of 1 °C/h. When the temperature reached 33 °C, the heat adhesion duration was recorded. 

### 2.5. Disease Resistance Evaluation 

#### 2.5.1. Bacterial Challenge Experiment

At the end of the feeding trial, 45 abalones (including 3 replicates, 15 abalones per replicate) were selected from each group for the evaluation of disease resistance. *Vibrio harveyi* AP37, at a lethal concentration of 50% (LC50, 1.0 × 10^7^ CFU/mL), was adopted in the bacterial challenge experiment. There was no feed for 168 h after being injected with 50 µL activated bacteria, and mortality was recorded at 9:00 a.m. and 6:00 p.m., respectively. The choice of *V. harveyi* AP37, the concentration determination, and activation method were all based on research by Zou et al. [33].

#### 2.5.2. Hemolymph Immune Response Determination

Another forty abalones per group were collected for further disease-resistance evaluation through hemolymph immune responses, and *V. harveyi* AP37 at a sublethal concentration (3.0 × 10^6^ CFU/mL) was adopted here. Among the selected abalones, thirty individuals were injected with 50 µL of *V. harveyi* AP37 suspension, and the remaining ten individuals were used as the negative control by being injected with an equal volume of sterile seawater. Three abalones per group were sampled each time at 0 (negative control group), 12, 24, 48, 72, and 96 h after injection, and 2 mL hemolymph of the individual was collected and kept on ice. Immune responses in the hemolymph were characterized using flow cytometry (CytoFLEX, Beckman Coulter, Indianapolis, IN, USA) by measuring the total hemocyte counts (THC) and mortality rate, phagocytic activity, and the production of reactive oxygen species (ROS). The determination methods for all hemolymph parameters referred to the protocols described by Zou et al. [33].

### 2.6. Intestinal Microbiome Analysis

Four abalones per group were used for total DNA extraction according to the CTAB method, and agarose gel electrophoresis was used to detect its concentration and purity. The genomic DNA was amplificated through PCR using the specific primers 515 F and 806 R. After mixing and purifying PCR products, a TruSeq^®^ DNA PCR-Free Sample Preparation Kit (Suzhou Renold Biotechnology, Suzhou, China) was used for the library construction. Finally, sequencing was performed using NovaSeq6000.

The raw data were quality-controlled using fastp to obtain clean data according to Bokulich et al. [34]. All effective data were then clustered using the Uparse software (v7.0.1001) and analyzed to determine their species annotation through the Mothur method using the SSUrRNA database from SILVA138. QIIME software (Ver1.9.1) was used to determine the alpha diversity index (goods coverage, PD whole tree, Shannon, Simpson, Chao1, and ACE), and to determine the beta diversity through the UniFrac distances for principal component analysis (PCA). Further, the LEfSe package in Python was utilized for linear discriminant analysis of the effect size (LEfSe), and functional annotation information was obtained using Tax4Fun; then, a cluster heatmap of functions in groups was also generated. In addition, to study the species with significant differences among groups, from the species abundance at different levels, the MetaStat method was used and the abundance distribution box chart was drawn. Other statistical analyses were performed in R software (v4.3.1).

### 2.7. Statistical Analysis

All data analyses were conducted using SPSS 20.0 (SPSS, Chicago, IL, USA). The results were presented as means ± standard error (SE). To investigate differences among groups, on the premise of passing the normality test and homogeneity-of-variance test, a one-way analysis of variance (ANOVA) was employed. When there were statistically significant differences, Duncan’s multiple-range test was used to compare the means of different groups. In addition, the heat adhesion duration and survival-rate curves were conducted with an overall comparison using the Log Rank Test. A *p* value < 0.05 was recognized as statistically significant.

## 3. Results

### 3.1. Survival and Growth Performance

After a 100-day feeding period, the dietary supplement of astaxanthin from the three sources all had obvious positive effects on the shell and weight growth (FSL, FW, DISL, WGR, and SGR) compared with the natural diet (GL group), but not in the survival rate, which all remained at a high level (>90%) (Table 2). In terms of weight-growth performance (FW, WGR, and SGR), there was a stronger improvement effect of dietary astaxanthin supplementation from the *Phaffia rhodozyma* and *Haematococcus pluvialis* sources (PR80 and HP120 group) compared with the industrial synthetic source (SA80 group). Significantly, the WGR in PR80 and HP120 groups grew by approximately 110% while the SA80 group also had about an 80% increase compared to the GL group. Also, the SGR of abalone in the PR80 and HP120 groups were boosted by nearly 75% compared to the GL group. As for the shell growth (FSL and DISL), the DISL in all dietary supplementation groups, in particular, rose to more than twice as much as the GL group. In addition, only the increase in the PR80 and HP120 groups reached significance in VIS compared with the GL group.

### 3.2. Serum Immune Index and Hepatopancreas Antioxidant Enzyme Activity

Four serum immune indexes of abalone were given in Table 3. The GLU contents of abalone from the three astaxanthin supplementation groups were all lower than that in the GL group, except for SA80, which did not reach a significant level (*p* > 0.05). The content of the COR of abalone from all supplementation groups was significantly lower compared with the GL group (*p* < 0.05). The AKP and LZM activity of abalone had the same trend, with all astaxanthin supplementation groups having significantly higher activity than the GL group.

As the results of the hepatopancreas antioxidant enzyme activity show in Table 3, there was significantly higher activities of T-AOC, CAT, SOD, and GSH-PX of abalone fed with dietary astaxanthin from three sources than those in the GL group, while the MDA content was significantly lower (*p* < 0.05). There was no significant difference between all the treatment groups (*p* > 0.05).

### 3.3. Heat-Resistance Evaluation in the Experiment

Figure 1 showed the results of the heat adhesion duration (HAD) of *H. discus hannai* fed diets with astaxanthin from different supplementation sources, under heat-resistance evaluation, during the experiment. The attachment rate of abalone in the GL group first reached half at about 4 h, and the maximum HAD was also the least among all the groups. The HADs of fifty percent attachment in the SA80 and HP120 groups were similar, and both were between 5 and 6 h. The attachment rate of abalone in PR80 dropped to 50% at approximately 6.5 h, and possessed the longest HAD at about 10 h.

### 3.4. Bacterial Challenge Experiment

The survival rate of abalone under the bacterial challenge experiment was shown in Figure 2. At 72 h after *V. harveyi* AP37 infection, the survival rates of abalone in all astaxanthin-supplementation groups were significantly higher than that in the GL group (below 50%). The survival rates in the PR80 and HP120 groups were both still above 75%, and were 82% in the PR80 group and 75% in HP120 group, respectively. The lowest survival rate was seen in the SA80 group among the three astaxanthin-supplementation groups. At the end of the trial, the correlation of the survival rate among all groups still stayed the same; PR80 maintained the top at about 66%, followed by the HP120 and SA80 groups.

### 3.5. Hemolymph Immune Responses

As shown in Figure 3, the THC of abalone in all groups had a similar changing trend under the bacterial challenge. At the beginning of the challenge (0 h), a significant difference was not witnessed in the THC of abalone in any of the groups (*p* > 0.05). At 12 h of the insertion, THC in all groups showed remarkable growth, but all supplementation groups presented a significantly higher number than the GL group (*p* < 0.05), among which the PR80 group achieved the top level. Up until 24 h after the challenge, all groups almost obtained peak values, and then showed a decrease with increasing challenge time. The relationship between groups did not change much; the THC in PR80 was always the maximum, followed by that of HP120 and then SA80. 

The results on hemocyte mortality were shown in Figure 4. Before the challenge, the hemocyte mortality of the three treatment groups was significantly lower than that of the GL group. With increasing infection time, the hemocyte mortality of abalone from all groups increased first and then experienced a drop but the relationship among groups did not change. The hemocyte mortality of astaxanthin supplementation groups always remained at a lower level compared with the GL group, with no remarkable difference among the treatment groups (*p* > 0.05).

As shown in Figure 5, at the start of the challenge, the ROS of all treatment groups were significantly higher than those of the control group (*p* < 0.05). After the challenge for 24 h, a surge in ROS in all groups started to appear, and those of the PR80 and HP120 groups rose more and were significantly higher than the SA80 and GL groups (*p* < 0.05). The ROS of abalone in the PR80 and HP120 groups then reduced from 48 h after the challenge until the three treatment groups showed no significant differences at the time of 96 h (*p* > 0.05), while they were still significantly higher than the control (*p* < 0.05).

The phagocytic activity of abalone under the bacterial challenge is shown in Figure 6. The changing tendency of phagocytic activity was similar to THC. There were no significant differences between any of the groups at first (*p* > 0.05). There was a significant gap in phagocytic activity of all the treatment groups from that of the GL group at the time of 12 h. Then, they all peaked at the time of 24 h, and then, a decline was witnessed at a later time. The phagocytic activity of the PR80 group always maintained the highest position since the increase began.

### 3.6. Intestinal Microbiome

#### 3.6.1. Intestinal Microbiota Diversity and Richness

The sequences were clustered into 576-3167 OTUs (Operational Taxonomic Units) for samples based on a 97% similarity level (Appendix A). As shown in Table 4, the goods coverage indices of the four groups were all >98%, which reflected the reliability of the results. The alpha diversity indices of the microbiota diversity and richness (Shannon, Simpson, Chao1, and ACE index) of the SA80 and PR80 groups were similar and had a higher value, while the HP120 and GL groups were also similar but remained at a lower level. In addition, the PD whole trees, which means the phylogenetic diversity, of the SA80 and PR80 groups were also higher than those of the HP120 and GL groups. Principal component analysis (PCA) based on the UniFrac distances was employed to determine the beta diversity; a clear separation was not seen but the GL and HP120 groups were closer, while the SA80 and PR80 groups were closer (Figure 7).

The relative abundances of the top 10 bacterial communities at the phylum level were illustrated in Figure 8A. Proteobacteria, Firmicutes, Fusobacteriota, Cyanobacteria, Actinobacteria, and Bacteroidota were the dominant microbes in all groups. The relative abundances of Firmicutes in the GL and HP120 groups (55.27% and 37.17%) were significantly higher than that of SA80 (28.30%) (*p* < 0.05), but not significantly higher than the PR80 group (33.03%). The relative abundances of proteobacteria in the GL and HP120 groups (20.20% and 21.87%) were significantly lower than those of the SA80 and PR80 groups (34.56% and 33.86%) (*p* < 0.05).

At the genus level, the relative abundances of the top 10 bacterial communities of different supplementation-source groups are shown in Figure 8B. *Mycoplasma*, *Psychrilyobacter*, and *Vibrio* also had the dominant relative abundances in all groups. Furthermore, unidentified_*Chloroplast*, *Ralstonia*, and *Burkholderia-Caballeronia-Paraburkholderia* dominated in the GL group, Serratia dominated in the HP120 group, and unidentified_*Chloroplast*, *Ralstonia* dominated in the SA80 group, whereas unidentified_*Chloroplast*, *Ralstonia* and *Serratia* dominated in the PR80 group. In addition, the relative abundances of *Mycoplasma* in the GL and HP120 groups (48.61% and 34.66%) were significantly higher than those of SA80 (13.08%) (*p* < 0.05). As for *Serratia* and *Vibrio*, the GL group had the least relative abundance (*p* < 0.05), 0.2% and 2.05%, respectively. The most abundant *Psychrilyobacter* was seen in the HP120 group (2.54%), and there was no significant difference among other groups (*p* > 0.05).

#### 3.6.2. LefSe Analysis and Function Clustering Heat Map of Intestinal Microbiota

As shown in Figure 9, compared to Group A, there were five biomarkers with significant differences in the GL group, seven biomarkers with significant differences in the HP120 group, only one biomarker with significant differences in the PR80 group, and twenty-three biomarkers with significant differences in SA80 group (*p* < 0.05).

Furthermore, these biomarkers in the groups were higher for genetic information processing, cell motility, signal transduction, neurodegenerative diseases, metabolism, etc., according to the function-prediction analysis (Figure 10). The clustering heat map also showed that four diet groups were segregated into two groups. One group was composed of the GL and HP120 groups, and the SA80 and PR80 groups were assigned to the other group.

#### 3.6.3. Significantly Different Species among Groups

Significantly different species of the intestinal microbiota of abalone fed diets with astaxanthin from different supplementation sources at different levels were found and are shown in Figure 11. In particular, the relative abundance of the PR80 group was significantly higher than that of the SA80 group. The relative abundance of the PR80 group being significantly higher than that of the SA80 group was found in Oscillospirales, Ruminococcaceae, and Staphylococcaceae in Firmicutes, and Yersiniaceae, *Serratia*, and Psedomonadaceae in Proteobacteria (*p* < 0.05).

## 4. Discussion

Aquatic animals are unable to synthesize carotenoids, so they need to be added into their feed [35,36]. Astaxanthin and other carotenoids have been widely used in aquatic feed formulations, achieving different biological functions [37]. Owing to the excellent performance of astaxanthin, studies involving this topic massively developed in the aquaculture area. Liu et al. [38] reported that 80 mg/kg astaxanthin can improve the antioxidative capability, increase the HSP70 level, and strengthen the crowding stress resistance of yellow catfish (*Pelteobagrus fulvidraco*). Regarding sea bass *Lateolabrax maculatus*, dietary supplementation of *H. pluvialis* improved the growth performance and antioxidant capacity of the fish. An increase was also witnessed in their immune response and bacterial resistance. Kheirabadi et al. [39] added red yeast as an astaxanthin source, and found that it could provide an optimal performance, antioxidant activity, and fillet pigmentation at a dose of 47 g/kg. As for crustaceans, Fawzy et al. [40] indicated that dietary astaxanthin in Pacific white shrimp *Litopenaeus vannamei* significantly improved its immune response; it could also increase its antioxidant capacity and resistance to white spot syndrome virus (WSSV). In our study, astaxanthin from industrial synthesis, *P. rhodozyma,* and *H. pluvialis* supplementation in the abalone diet performed better in all aspects, including survival, growth, immune response, antioxidant activity, thermal resistance, and disease resistance, compared with common algae feeding. This is quite similar to previous studies on other aquatic animals. 

Although there were some differences among them because of the supplementation level and the sources of supplementation, these results contribute to improving our understanding of the addition of astaxanthin in aquaculture. Certainly, some researchers have noticed the problem of the sources of supplementation, like our study. For instance, Su et al. [41] compared the effect of dietary supplementation with *H. pluvialis* and synthetic astaxanthin on carotenoid composition, concentration, esterification degree, and astaxanthin isomers in the ovaries, hepatopancreas, carapace, and the epithelium of Chinese mitten crab (*Eriocheir sinensis*), and the results suggest that *H. pluvialis* is more effective in astaxanthin accumulation, and it is better at increasing the carotenoid and astaxanthin isomer composition compared with synthetic astaxanthin. Xie et al. [42] compared the effects of synthetic astaxanthin and *H. pluvialis* supplementation in golden pompano (*Trachinotus ovatus*). Astaxanthin supplementation from two sources all provided the promotion of growth performance, antioxidant capacity, and anti-inflammation, but there were no significant differences between them. The effects of natural astaxanthin from *H. pluvialis* and synthetic astaxanthin supplementation were also evaluated on the ridgetail white prawn (*Exopalaemon carinicauda*). The results illustrated that natural astaxanthin from *H. pluvialis* showed superiority in growth performance, pigmentation, and astaxanthin content than that from synthetic astaxanthin. In our study, as for survival and growth performance, astaxanthin from the two natural sources caused better growth performances in abalone than the industrial synthetic source. In terms of heat stress and bacterial challenge, abalone fed dietary astaxanthin from *Phaffia rhodozyma* showed the strongest resistance. In brief, the supplementation effects of different astaxanthin sources were different, even among the natural sources.

It is clear that astaxanthin dietary supplementation significantly altered the gut microbiota structure of abalone according to the alpha diversity and beta diversity data in this study. However, when relying on clustering results and diversity data, the fact that the intestinal microbiota structure of the HP120 group was not affected too much and was similar to that of the GL group may look confusing. The intestinal microbiota structure of SA80 and PR80 changed a lot and influenced the level and manner to a similar extent, clustering into a group in clustering analysis. The variation in the intestinal microbiota structure cannot match the results of other production performances like growth, stress resistance, disease resistance, and immunity. The intestinal microbiota structure of the HP120 group was similar to the GL group, but the three supplementation groups all achieved better performances compared to the GL group. It is highly possible that HP120 achieved positive effects through other approaches, so we focused on the other two astaxanthin supplementation groups in which the intestinal microbiota may have made greater contributions. Hence, we discovered all the significantly different species of the intestinal microbiota of abalone among groups at different levels. The situation that the relative abundance of the PR80 group was significantly higher than that of the SA80 group was especially notable because the effects of the PR80 group were better than the SA80 group. 

According to our results, the relative abundance in the PR80 group being significantly higher than that in the SA80 group was found in Oscillospirales, Ruminococcaceae, and Staphylococcaceae in Firmicutes and Yersiniaceae, *Serratia*, and Psedomonadaceae in Proteobacteria. Proteobacteria and Firmicutes were the dominant phylum in most aquatic animals’ intestines [43,44,45]. Proteobacteria are a group of Gram-negative bacteria, whose functions are associated with the nutrient cycle (carbohydrate fermenting) and the mineralization of organic compounds [46,47]. A recent study from Yu et al. [48] also reported that pacific abalone with a higher feed efficiency (FE) had more abundant Proteobacteria. Therefore, in this study, *P. rhodozyma* supplementation (80 mg/kg astaxanthin) in the abalone diet obtained a better production performance because of the high percentage of intestinal Proteobacteria, which had a stronger ability to utilize carbohydrates and an increased FE than the added synthetic astaxanthin in the abalone diet. A significantly higher relative abundance of Firmicutes was also observed in the PR80 group. Some studies have evidenced that a higher relative abundance of Firmicutes means a higher energy-harvesting ability, specifically reflected in a higher fatty-acid-absorption and lipid-metabolism ability [49,50,51]. In this study, dietary astaxanthin in the source of *P. rhodozyma* supplementation caused a higher abundance of Firmicutes, and the digestion and metabolism of lipids in the host were strengthened, finally resulting in a better growth performance. 

In detail, *Serratia*, Yersiniaceae, and Psedomonadaceae belonging to Proteobacteria showed a higher abundance in the PR80 group compared to the synthetic astaxanthin supplementation group. There has been limited research to understand the function and effect of *Serratia* in animals’ gut. Present acknowledgements about *Serratia* have been divergent: some studies have held the view that *Serratia* is a Gram-negative opportunistic pathogen, and can injure intestinal epithelial cells causing gastrointestinal problems, and has consequences for the host’s physiology and health [52,53,54]. On the contrary, Bai et al. [55] illustrated that *Serratia* can activate innate immune responses of *Drosophila* through Toll and Imd immune-signaling pathways to achieve the goal of anti-*Plasmodium* and antibacterial immune defense. Also, the study of Nie et al. [56] proved that *Serratia* can produce a bioactive substance Prodigiosin (PG), a red microbial pigment with a tetrapyrrole ring structure. It has multiple biological functions, including anticancer, antimalarial, antibacterial, and immunomodulatory functions. Therefore, combining the results of this experiment, the absolute predominance of *Serratia* in the PR80 group compared to the SA80 group should result in a positive influence on abalone, such as representing antibacterial, and immunomodulatory function, so that advantages can be seen in the heat-resistance evaluation experiment and the bacterial challenge experiment. The significantly different species Psedomonadaceae also belongs to Proteobacteria. In terms of insect biology, gut Pseudomonas in olive fly guts can help hydrolyze the proteins of olive flesh [57]. This represents its ability of plant protein digestion. Furthermore, Pseudomonas are commonly found and investigated in marine invertebrates, especially sponge. Some species of Pseudomonas have been isolated and proven to possess antimicrobial activity against one or more pathogens, and can be potential candidates for broad-spectrum antibiotics [58,59]. As a consequence, a higher abundance of Pseudomonas in the intestines can play a more positive role in antibiosis and plant protein digestion, so that it can adopt a more proactive immune response to resist bacterial challenge and make an improvement in growth.

As for another significantly different species Firmicutes, Ruminococcaceae participates in the degradation and fermentation of polysaccharides and fibers [60]. Its main metabolites, short-chain fatty acids (SCFAs), are related to multiple functions [61], such as providing energy [62], anti-inflammatory and anticancer properties [63,64], improving the intestinal epithelial barrier function [65], antibiosis [66] and stimulating hormone secretion [67]. Staphylococcaceae is recognized as a pathogenic bacteria and is associated with the systemic inflammation of a host [68,69]. Although the most abundant ‘harmful bacteria’ Staphylococcaceae was seen in the HP120 group, no adverse impact appeared because of the low level of relative abundance. In general, Firmicutes, with the highest Ruminococcaceae as its characteristic, benefited abalone’s growth and abilities of immune and environmental stress resistance when abalone were fed with the diet consisting of *P. rhodozyma* astaxanthin. 

The intestinal microbe Lactococcus also cannot be ignored; related results are shown in the LEfSe analysis. Lactococcus was the only biomarker in the PR80 group, which may help to explain the reason why comprehensive advantages were presented in the PR80 group. According to extensive reports, Lactococcus is a marker of the health of the intestines, and can promote the recovery of blood metabolites [70] and the inhibition of pathogen planting and growth [71,72]. Thus, Lactococcus resulted in better intestinal health in the PR80 group and finally contributed to resistance to disease and heat, as well as immune function.

Astaxanthin supplementation from different sources has been proved to impose influence on abalone by different pathways and approaches; however, what caused the addition of the ‘same’ astaxanthin to cause different results? Astaxanthin has three configurational isomers, including stereoisomers (enantiomers, (3S, 3′S) and (3R, 3′R)) and a geometric isomer (meso form, (3R, 3′S)). There are different isomers contained in different astaxanthin sources. *H. pluvialis* only possesses the (3S, 3′S) isomer, while yeast *P. rhodozyma* has the (3R, 3′R) isomer. However, all three types of optical isomers can be found in synthetic astaxanthin, at a proportion of 1:2:1 of (3S, 3′S) (3R, 3′S) and (3R, 3′R) isomers [30,73,74].

In addition, astaxanthin exists in free and esterified forms, because it can be esterified in hydroxyl groups with different fatty acids or without esterification or combined with proteins [75]. Interestingly, astaxanthin in algae is always esterified and it is on the opposite side in synthetic astaxanthin [76]. It has been reported that esterified astaxanthin from microalgae has a poorer performance on coloration in rainbow trout than unesterified astaxanthin [77]. However, some studies like Capelli et al. [78] found that algal-based astaxanthin is superior to synthetic astaxanthin regarding antioxidant appearance.

Therefore, the best form of astaxanthin should depend on the demand: it needs to be accurately, efficiently, and sustainably produced in the service industry. In brief, isomer differences and their associations with compounds may be the cause of production performance differences (growth, stress resistance, disease resistance, and intestinal microbiota community structure).

## 5. Conclusions

The supplementation of astaxanthin from industrial synthesis, *Phaffia rhodozyma*, and *Haematococcus pluvialis* into the abalone diet performed better in all aspects including survival, growth, immune response, antioxidant activity, thermal resistance, and disease resistance, compared with common algae feeding. As for survival and growth performance, astaxanthin sources from *Phaffia rhodozyma* and *Haematococcus pluvialis* had better growth performances than the industrial synthetic source. In terms of heat stress and bacterial challenge, abalone fed a diet of dietary astaxanthin from the *Phaffia rhodozyma* source showed the strongest resistance. Overall, 80 mg/kg astaxanthin supplementation from *Phaffia rhodozyma* is recommended to achieve a more effective and comprehensive outcome.

## Figures and Tables

**Figure 1 antioxidants-12-01641-f001:**
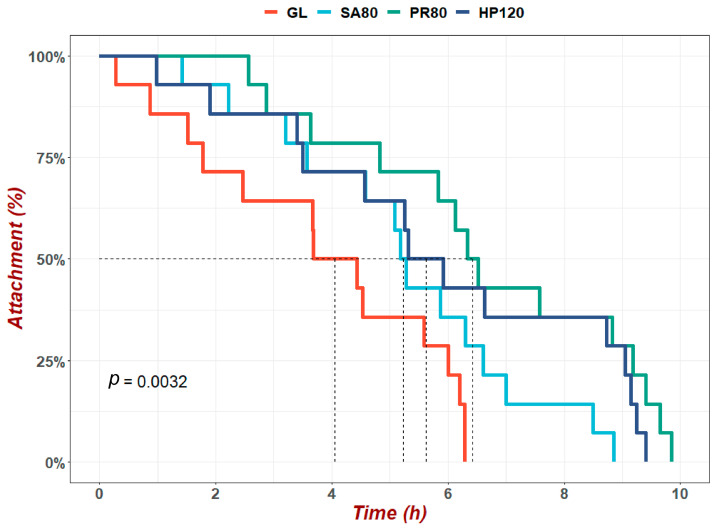
Heat adhesion duration (HAD) of *H. discus hannai* fed diets with astaxanthin from different supplementation sources under heat-resistance evaluation. The values marked by dashed lines were the time when the attachment rate dropped to a half.

**Figure 2 antioxidants-12-01641-f002:**
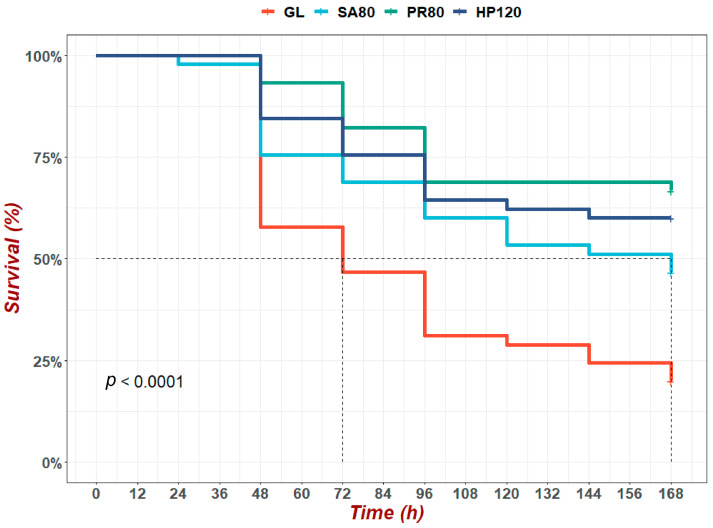
The survival rate of *H. discus hannai* fed diets with astaxanthin from different supplementation sources under the bacterial challenge experiment. The values marked by dashed lines were the time when the survival rate dropped to a half.

**Figure 3 antioxidants-12-01641-f003:**
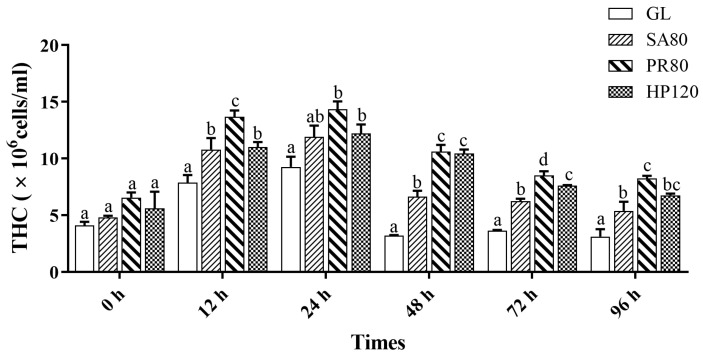
Total hemocyte counts (THC) of *H. discus hannai* fed diets with astaxanthin from different supplementation sources under the bacterial challenge. Different letters in different groups indicate significant differences (*p* < 0.05).

**Figure 4 antioxidants-12-01641-f004:**
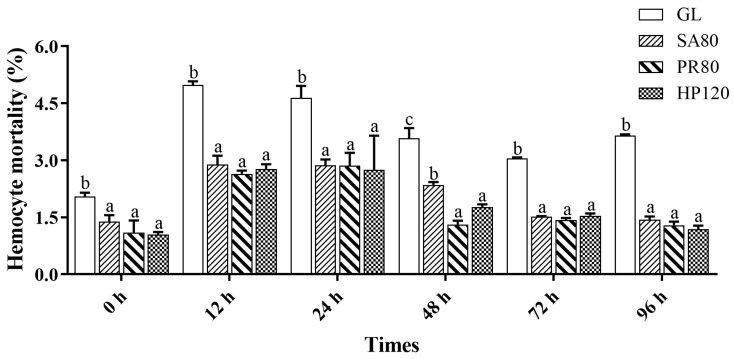
The hemocyte mortality rate of *H. discus hannai* fed diets with astaxanthin from different supplementation sources under bacterial challenge. Different letters in different groups indicate significant differences (*p* < 0.05).

**Figure 5 antioxidants-12-01641-f005:**
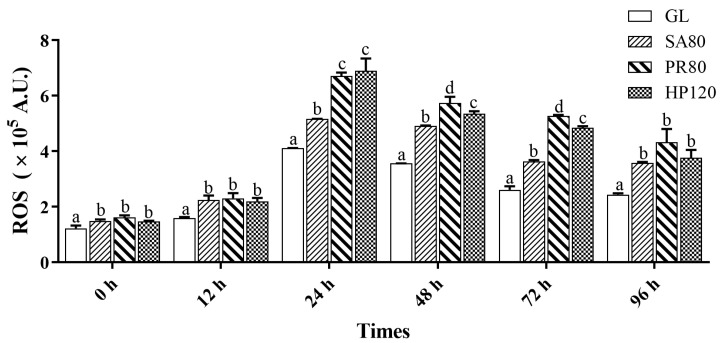
Reactive oxygen species (ROS) of *H. discus hannai* fed diets with astaxanthin from different supplementation sources under bacterial challenge. Different letters in different groups indicate significant differences (*p* < 0.05).

**Figure 6 antioxidants-12-01641-f006:**
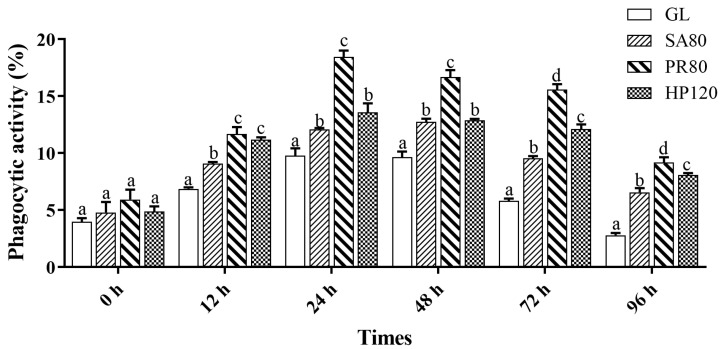
Phagocytic activity of *H. discus hannai* fed diets with astaxanthin from different supplementation sources under bacterial challenge. Different letters in different groups indicate significant differences (*p* < 0.05).

**Figure 7 antioxidants-12-01641-f007:**
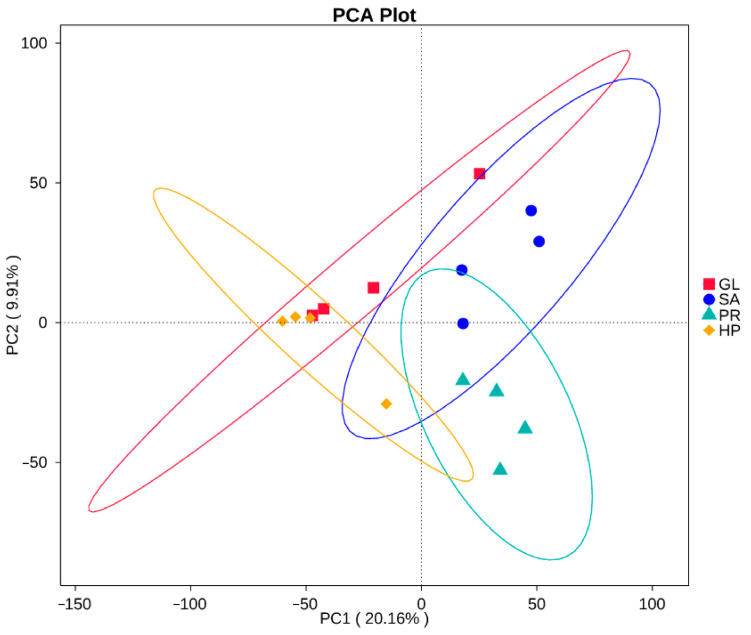
Beta diversity of the intestinal microbiota of *Haliotis discus hannai* fed diets with astaxanthin from different supplementation sources. Analyzed by principal component analysis (PCA) using PC1 versus PC2 axes based on UniFrac distance.

**Figure 8 antioxidants-12-01641-f008:**
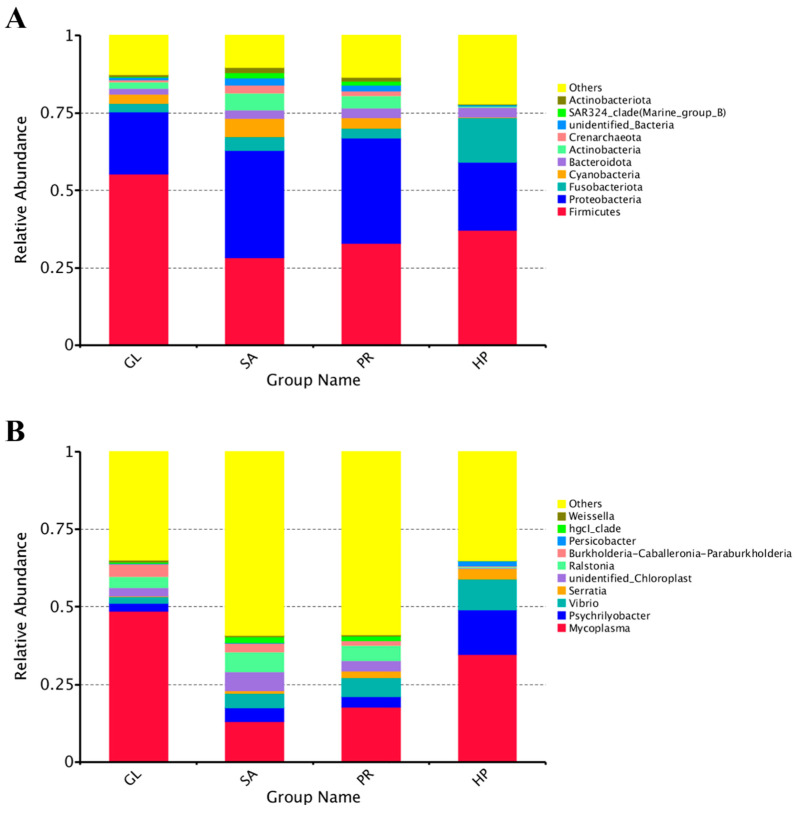
Relative abundance of the top 10 intestinal microbiota of *Haliotis discus hannai* fed diets with astaxanthin from different supplementation sources at phylum (**A**) and genus (**B**) level.

**Figure 9 antioxidants-12-01641-f009:**
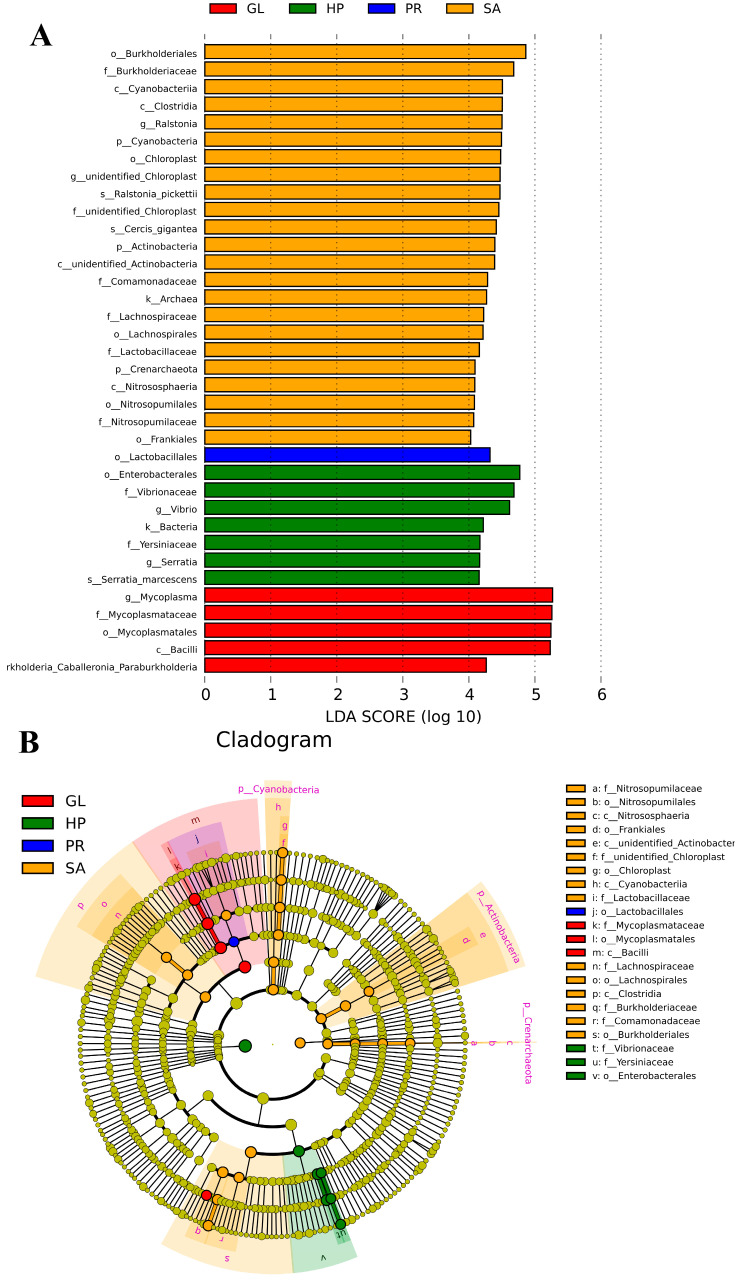
Linear discriminant analysis of effect size (LEfSe) of intestinal microbiota of *Haliotis discus hannai* fed diets with astaxanthin from different supplementation sources. (**A**) Linear discriminant analysis scores of the abundance of taxa. (**B**) Cladogram showing differences in the abundance of taxa.

**Figure 10 antioxidants-12-01641-f010:**
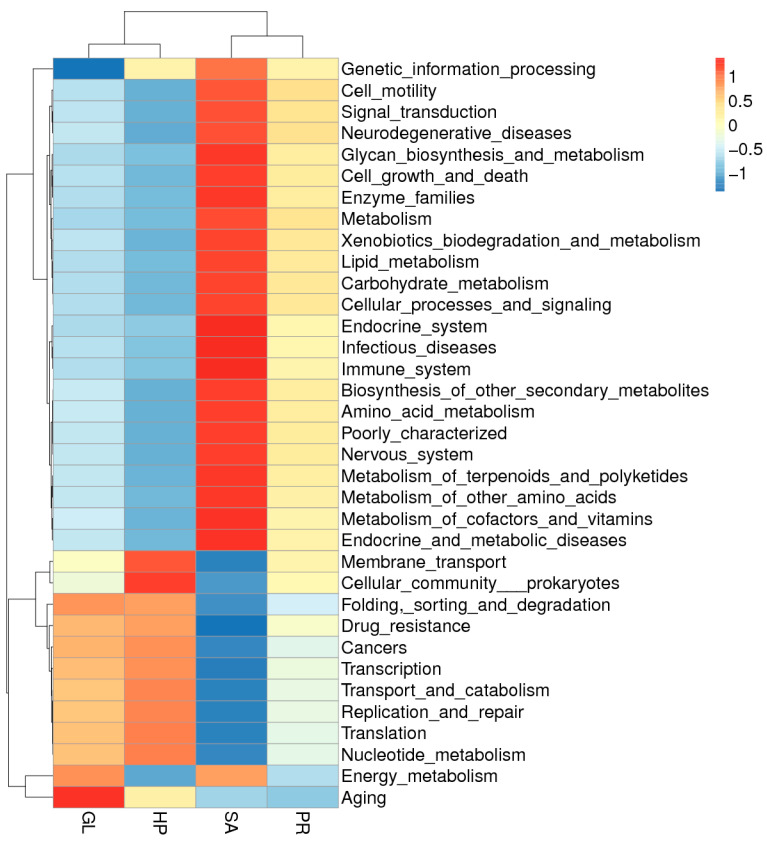
The clustering heat map analysis of the function at level 2 of the intestinal microbiota of *Haliotis discus hannai* fed diets with astaxanthin from different supplementation sources.

**Figure 11 antioxidants-12-01641-f011:**
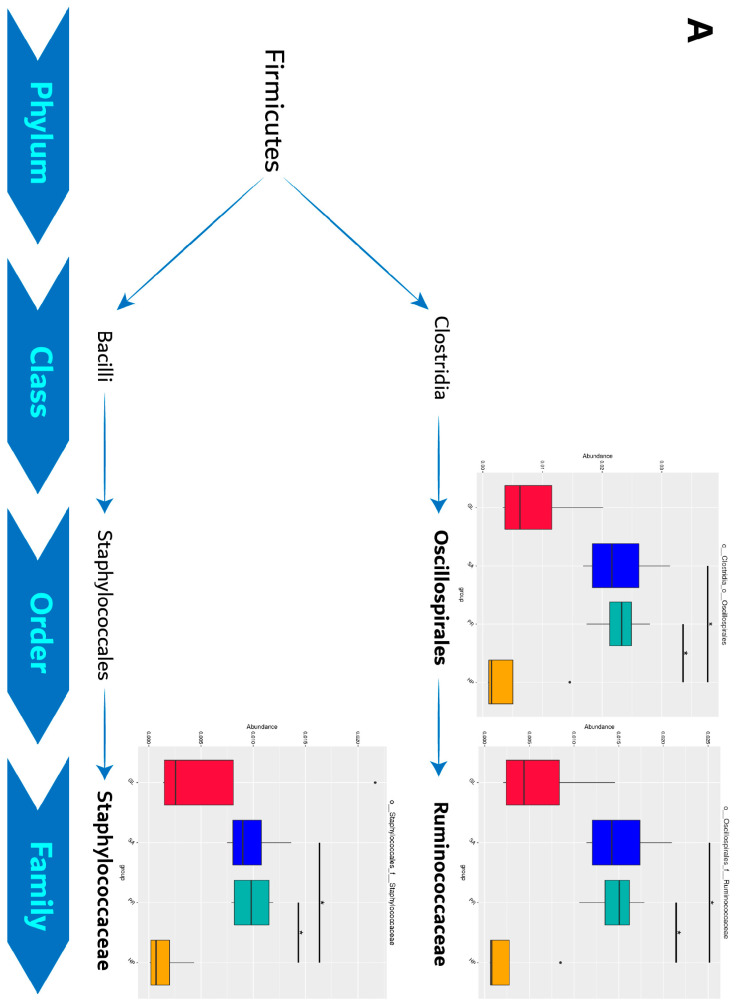
Analysis of species with differences among groups (the relative abundance of the PR80 group especially was significantly higher than that of the SA80 group). (**A**) Species with differences in Firmicutes; (**B**) Species with differences in Proteobacteria. Asterisk among different groups means significant differences (* *p* < 0.05, ** *p* < 0.01).

**Table 1 antioxidants-12-01641-t001:** Formulation and proximate composition of the experimental diets (dry-matter basis, g/kg).

Parameters	Experimental Diets			
	GL	SA80	PR80	HP120
Fish meal ^a^		150	150	150
Wheat gluten ^a^		120	120	120
Soybean meal ^a^		130	130	130
High-gluten flour ^a^		129.2	110	126
Kelp powder ^a^		250	250	250
Dextrin ^b^		50	50	50
α-starch ^b^		50	50	50
Sodium alginate ^c^		30	30	30
Soybean lecithin ^d^		20	20	20
Cholesterol ^e^		5	5	5
Vitamin premix ^f^		20	20	20
Mineral premix ^g^		30	30	30
Choline chloride		5	5	5
Monocalcium phosphate		5	5	5
Astaxanthin ^h^		0.8		
*Phaffia rhodozyma* ^i^			20	
*Haematococcus pluvialis* ^j^				4
Proximate composition (%)				
Moisture	82.48 ± 0.55	5.26 ± 0.57	5.40 ± 0.30	6.40 ± 0.13
Crude protein	16.37 ± 0.28	31.80 ± 0.20	32.97 ± 0.76	32.10 ± 0.38
Crude lipid	1.86 ± 0.17	3.83 ± 0.40	3.67 ± 0.19	3.73 ± 0.15
Ash	21.13 ± 0.17	18.64 ± 0.26	17.97 ± 0.22	18.15 ± 0.03

GL group: natural diet *Gracilaria lemaneiformis*; SA80 group: 800 mg/kg industrial synthetic astaxanthin supplementation (80 mg/kg astaxanthin actual weight); PR80 group: 20 g/kg *Phaffia rhodozyma* supplementation (80 mg/kg astaxanthin actual weight); HP120 group: 4 g/kg wall-broken *Haematococcus pluvialis* dry powder supplementation (120 mg/kg astaxanthin actual weight). ^a^ Provided by Fuzhou Promarine Biotechnology Co., Ltd. (Fuzhou, China). ^b^ Purchased from Zhengzhou Kangyuan Chemical Products Co., Ltd. (Zhengzhou, China). ^c^ Purchased from Qingdao Haizhilin Biotechnology Development Co., Ltd. (Qingdao, China). ^d^ Soybean lecithin had 51.5% PC, 23.3% PE, 10% PI and trace amounts of PS, and 96.9% acetone insolubility. Purchased from Shanghai Taiwei, Co., Ltd. (Shanghai, China). ^e^ Cholesterol: effective content ≥ 95%. Purchased from Shanghai Macklin Biochemical Co., Ltd. (Shanghai, China). ^f^ Vitamin mixture (IU or g/kg mixture): Vitamin A, 100,000 IU; Vitamin D3, 25,000 IU; Vitamin E, 4000 IU; Vitamin K3, 0.45 g; Vitamin B1, 1 g; Vitamin B2, 1 g; Vitamin B3, 0.8 g; Vitamin B6, 1 g; Vitamin B12, 0.005 g; Vitamin C, 4 g; Folic acid, 0.45 g; Pantothenic acid, 3.5 g; Nicotinic acid, 7 g; Inositol, 8 g; Biotin, 0.05 g; L-carnitine, 0.25 g; 4-aminobenzoic, 0.4 g. ^g^ Mineral premix (g/kg mixture): CuSO_4_, 0.32 g; MnSO_4_, 0.7 g; ZnSO_4_, 3.2 g; FeSO_4_, 8 g; MgSO_4_, 3 g; KIO_3_, 0.065 g; Na_2_ SeO_3_, 0.025 g; CoSO_4_, 0.060 g. ^h^ Carophyll pink from DSM Nutritional Products France, 10% astaxanthin. ^i^ Astaxanthin content of dry matter: 0.4%. Purchased from Fujian Lifecome Biochemistry Co., Ltd. (Nanping, China). ^j^ Astaxanthin content of dry matter 3%, Yunnan Kunming Biogenic Co., Ltd. (Kunming, China).

**Table 2 antioxidants-12-01641-t002:** Survival and growth performance of *H. discus hannai* fed diets of astaxanthin from different supplementation sources.

Parameters	Experimental Diets			
	GL	SA80	PR80	HP120
SR (%)	91.67 ± 1.67 ^a^	92.50 ± 1.12 ^a^	93.33 ± 1.05 ^a^	93.33 ± 2.47 ^a^
FSL (cm)	64.60 ± 0.93 ^a^	73.47 ± 0.49 ^b^	74.17 ± 0.83 ^b^	74.16 ± 0.78 ^b^
FW (g)	35.62 ± 0.92 ^a^	46.56 ± 0.90 ^b^	48.92 ± 0.57 ^c^	48.81 ± 0.33 ^c^
DISL (µm/day)	57.71 ± 9.48 ^a^	135.01 ± 3.93 ^b^	141.79 ± 9.05 ^b^	146.00 ± 8.36 ^b^
WGR (%)	46.25 ± 3.65 ^a^	85.38 ± 3.82 ^b^	97.62 ± 3.35 ^c^	95.98 ± 1.77 ^c^
SGR (%)	0.42 ± 0.03 ^a^	0.68 ± 0.02 ^b^	0.76 ± 0.02 ^c^	0.75 ± 0.01 ^c^
VIS (%)	15.85 ± 0.52 ^a^	17.20 ± 0.27 ^ab^	18.11 ± 0.56 ^b^	18.17 ± 0.59 ^b^

FSL: final shell length; FW: final weight; SR: survival rate; DISL: daily increment in shell length; WGR: weight gain rate; SGR: specific growth rate; VIS: visceral–somatic index. Different superscript letters indicate statistical significance.

**Table 3 antioxidants-12-01641-t003:** Serum immune index and hepatopancreas antioxidant enzyme activity of *H. discus hannai* fed diets with astaxanthin from different supplementation sources.

Parameters	Experimental Diets			
	GL	SA80	PR80	HP120
**Serum**				
GLU (mmol/L)	0.23 ± 0.02 ^b^	0.19 ± 0.01 ^ab^	0.16 ± 0.01 ^a^	0.15 ± 0.01 ^a^
COR (ng/mL)	40.11 ± 1.47 ^b^	32.49 ± 1.15 ^a^	33.36 ± 2.02 ^a^	31.5 ± 0.96 ^a^
AKP (U/L)	224.9 ± 3.42 ^a^	241.67 ± 4.97 ^b^	245.88 ± 3.87 ^b^	243.52 ± 1.59 ^b^
LZM (µg/mL)	3.96 ± 0.28 ^a^	5.36 ± 0.17 ^b^	5.44 ± 0.31 ^b^	5.33 ± 0.18 ^b^
**Hepatopancreas**				
T-AOC (U/mg prot)	0.39 ± 0.03 ^a^	0.61 ± 0.04 ^b^	0.68 ± 0.03 ^b^	0.68 ± 0.05 ^b^
CAT (U/mg prot)	81.86 ± 1.91 ^a^	117.99 ± 3.90 ^b^	124.70 ± 1.66 ^b^	126.34 ± 4.84 ^b^
SOD (U/mg prot)	16.18 ± 0.94 ^a^	20.15 ± 0.32 ^b^	20.42 ± 0.21 ^b^	20.74 ± 0.90 ^b^
GSH-PX (U/mg prot)	85.99 ± 6.02 ^a^	106.86 ± 4.53 ^b^	122.67 ± 3.93 ^b^	117.70 ± 6.16 ^b^
MDA (nmol/mg prot)	8.43 ± 0.43 ^b^	4.40 ± 0.21 ^a^	3.83 ± 0.10 ^a^	3.82 ± 0.25 ^a^

All dates are presented as means ± standard error (SE) of the mean (n = 6). Values with different superscript letters within the same line were significantly different (*p* < 0.05).

**Table 4 antioxidants-12-01641-t004:** Alpha diversity indices of the intestinal microbiota of *Haliotis discus hannai* fed diets with astaxanthin from different supplementation sources.

Groups	Goods Coverage	PD Whole Tree	Shannon	Simpson	Chao1	ACE
GL	0.993	171.907	4.741	0.751	1975.896	2041.389
SA80	0.99	253.411	7.811	0.972	3236.198	3305.304
PR80	0.989	266.931	7.779	0.969	3432.391	3487.38
HP120	0.996	115.323	4.354	0.869	1173.165	1210.052

## Data Availability

The data presented in this study are available on request from the corresponding author.

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
