# Peer review of "Comprehensive Comparison of Effects of Antioxidant (Astaxanthin) Supplementation from Different Sources in Haliotis discus hannai Diet"

_antioxidants, 2023, doi:10.3390/antiox12081641_

Round 1

Reviewer 1 Report

This Ms addresses a relevant (but not original) topic in aquaculture and may provide baseline information of the culture of the species considered in this investigation. The Introduction provides an adequate state-of-the art on the research topic and the main questions addressed by the research are sufficiently identified.

The methods are those used in similar investigation on this topic and are well and sufficiently described. The statistical analysis is also standard-based and adequate for exploring the results/data-set and support the underlying scientific questions.

The Results section is well organised and the results sufficiently described.

The Discussion provides a good integration of the results which are sufficiently analysed. Maybe the Discussion section could be a bit condensed (i.e., shortened) without loosing the focus (and output/contribution) of the overall section.

The list of references is correctly organized, in terms of layout and includes the relevant support information for the Ms.

In spite of my condition of non-English native (expressed somewhere in this evaluation) I feel that the Ms needs some improvements in the quality of the English.

This is may main suggestion for revision as for the rest the Ms is well presented and, overall, reads well.

As mentioned, this a traditional approach in this kind of investigation and thus the originality is insufficiently demonstrated but there is some contribution to improving the culture of the species considered in the investigation.

Author Response

Reply:  Thank you so much for your kind comments and suggestions. The Discussion section has been put under a bit condensed. Besides, we will keep learning to make progress in future writing. The manuscript will be sent for professional language editing before publishment to improve English description.

Reviewer 2 Report

Please, at the end of Introduction re-write the aim of the research because it is not reported the different sources of pigment used to perform the study.

Author Response

Please, at the end of Introduction re-write the aim of the research because it is not reported the different sources of pigment used to perform the study.

Reply: Thank you so much for your generous advices.  The aim of the research at the end of Introduction has been re-written. [Line 85-94]

Reviewer 3 Report

Review for the paper "Comprehensive effects comparison of antioxidant (astaxanthin) supplementation from different sources in Haliotis discus hannai diet" by Weiguang Zou, Jiawei Hong, Wenchao Yu, Yaobin Ma, Jiacheng Gan, Yanbo Liu, Xuan Luo , Weiwei You, Caihuan Ke submitted to "Antioxidants".

General comment.

The authors conducted an experimental study to determine the effects of astaxanthin supplementation in the diet of the abalone Haliotis discus hannai on growth performance, survival, immune response, antioxidant activity, and thermal and disease resistance. The authors found that the source of astaxanthin was important, as different sources produced different results. The authors concluded that supplementation with 80 mg/kg astaxanthin from the source Phaffia rhodozyma showed the best results and can be recommended to the industry. These findings may have significant implications for abalone aquaculture. The authors used standard methods for abalone rearing, sample collection and processing, and statistical treatment of data.

Recommendations:

The abstract is not informative because the authors do not provide definitions for the different experimental groups.

Table 1. The authors should update the footnote to explain the abbreviations used to identify the diet groups.

For the statistical analysis, the authors used a parametric approach, specifically one-way ANOVA. However, the prerequisites for using this method - data with normal distribution and homogeneity of variances - were not addressed in the text. Authors are strongly urged to examine their data for normality and heteroscedasticity, to apply transformations if necessary, or to use nonparametric methods if appropriate.

The authors did not provide information on the statistical method they used to compare the duration of heat adhesion (Fig. 1) and survival rates during the bacterial challenge experiment (Fig. 2) among the experimental groups.

Figure 9 is difficult to understand because the font size is too small.

In general, the results of this study appear promising, but in its current form this paper is not acceptable because the English does not meet the standards of peer-reviewed journals.

The English must be improved.

Author Response

The abstract is not informative because the authors do not provide definitions for the different experimental groups.

Reply: Thank you so much for your advices. The definitions for the different experimental groups have been added. [Line19-22]

Table 1. The authors should update the footnote to explain the abbreviations used to identify the diet groups.

Reply: Thank you so much for your advices. The footnote to explain the abbreviations used to identify the diet groups has been updated. [Line 139-142]

For the statistical analysis, the authors used a parametric approach, specifically one-way ANOVA. However, the prerequisites for using this method - data with normal distribution and homogeneity of variances - were not addressed in the text. Authors are strongly urged to examine their data for normality and heteroscedasticity, to apply transformations if necessary, or to use nonparametric methods if appropriate.

Reply: We have done the normality test and variance homogeneity test before one-way ANOVA. For a concise description, we have omitted this section. All analysis using one-way ANOVA passed the normality test and variance homogeneity test. Supplementary description has been added. [Line 251]

The authors did not provide information on the statistical method they used to compare the duration of heat adhesion (Fig. 1) and survival rates during the bacterial challenge experiment (Fig. 2) among the experimental groups.

Reply: We are so sorry about the information lack. Related information has been added in Materials and Methods part. [Line254-255]

Figure 9 is difficult to understand because the font size is too small.

Reply: Thank you so much for your advices. Figure layout has been changed to make the figure bigger for clear read.

In general, the results of this study appear promising, but in its current form this paper is not acceptable because the English does not meet the standards of peer-reviewed journals.

Reply: We are so sorry about our English writing, we will keep learning to make progress in future writing. The manuscript will be sent for professional language editing before publishment to improve English description.

Round 2

Reviewer 3 Report

The authors have addressed my concenrs. 

The English should be revised.

Author Response

Thank you very much for your time and all your efforts. The manuscript will be edited and the revision submitted.